# Synergistic Effect of Glycyrrhizic Acid and ZnO/Palygorskite on Improving Chitosan-Based Films and Their Potential Application in Wound Healing

**DOI:** 10.3390/polym13223878

**Published:** 2021-11-10

**Authors:** Qian Zhang, Hong Zhang, Aiping Hui, Junjie Ding, Xinyue Liu, Aiqin Wang

**Affiliations:** 1Laboratory Medicine Center, Lanzhou University Second Hospital, Lanzhou 730030, China; qzhang2019@lzu.edu.cn; 2Key Laboratory of Clay Mineral Applied Research of Gansu Province, Center of Eco-Material and Green Chemistry, Lanzhou Institute of Chemical Physics, Chinese Academy of Sciences, Lanzhou 730000, China; zhanghong@licp.cas.cn (H.Z.); aphui1215@163.com (A.H.); 17693225101@163.com (J.D.); 3Center of Materials Science and Optoelectronics Engineering, University of Chinese Academy of Sciences, Beijing 100049, China

**Keywords:** chitosan, glycyrrhizic acid, ZnO/PAL, antibacterial, nanocomposite films

## Abstract

The synergistic effect of chitosan (CS), glycyrrhizic acid (GA) and ZnO/palygorskite (ZnO/PAL) as potential wound dressing was evaluated in the form of films by the solution casting method. The nanocomposite films were well-characterized with ATR-FTIR, XRD and SEM to explore the interactions between CS, GA and ZnO/PAL. Physical, mechanical and antibacterial properties of the nanocomposite films were systematically investigated for their reliability in end-up utilization. Importantly, it was found that the presence of PAL in the films provided enhanced mechanical properties, whereas CS, GA and ZnO supplied a broad-spectrum antibacterial activity, especially for drug-resistant bacteria such as ESBL—*E. coli* and MRSA. Overall, this research demonstrated that the prepared films can be a promising candidate for wound-care materials.

## 1. Introduction

The overuse of antibiotics has led to the emergence of drug-resistant bacteria and then the spread of drug-resistant bacterial infections. Gram-negative bacteria *Escherichia coli* (*E. coli*) and Gram-positive bacteria *Staphylococcus aureus* (*S. aureus*) are the two most common pathogenic bacteria, and in addition to the above two, the infection caused by extended-spectrum β-lactamases-producing *Escherichia coli* (ESBL—*E. coli*) or methicillin-resistant *Staphylococcus aureus* (MRSA) would lead to higher morbidity and mortality rates. All of the pathogenic bacteria are seriously threatening the health of human beings [1,2], and thus finding high-efficient and broad-spectrum tactics against drug-resistant bacteria is becoming very important [1,3]. Skin plays a crucial role in keeping bacteria out for mankind, but meanwhile it is extremely fragile and easily damaged, which plays right into the hands of bacterial [4]. Infected skin may prevent wounds from quickly and completely healing and may even lead to a battery of serious complications [5]. Hence, effective anti-infection control aiming to accelerate wound healing has caught a growing number of scholars’ attention [6], and it is particularly vital to exploit wound dressings with multiple anti-infective capabilities to cope with the latent jeopardy of fearsome drug-resistant bacteria. In this area, biological macromolecules based hydrogels and films gradually show their unique advantages, especially those with antibacterial properties [7,8].

Chitosan (CS) is the second most abundant natural biopolymer which is derived from the partial deacetylation of chitin, which is usually available in cephalopod species [9]. CS has been widely used as wound dressings in medicine and pharmaceutical fields due to its splendid biocompatibility, distinct antibacterial activity and nontoxicity, where CS plays an important role in inhibiting the infections and facilitating the recovery of the wounds, which can be attributed to its positive charge given by amino groups [10,11,12,13]. Additionally, with the joint efforts of numerous researchers, CS-based materials have made great progress in the above areas [14]. However, though a lot of CS-based films have been developed, the disappointing mechanical properties and insufficient antibacterial performance still impose restrictions on their end-use applications [15,16]. Therefore, how to find an efficient and economical way to improve both defects of the films still needs to be researched.

A variety of fillers have been incorporated into the CS matrix to improve the overall performance of the films [17], and clay minerals can be the gifted filler due to their environmental friendliness, abundant reserves and other advantages [18]. Different from lamellar clay minerals (i.e., montmorillonite, kaolinite, illite), palygorskite (PAL) has a unique sandwich-shaped structural unit, and eventually forms a special rod-like crystal structure and also zeolite-like channels and surface active silanol groups, which endow PAL with excellent supporting properties (rods), carrier properties (channels) and reactivity (silanol groups) [19,20]. ZnO nanoparticles (ZnO NPs) have attracted attention in enhancing the biological functions of nanocomposites for their extraordinary antibacterial activity [21,22,23,24,25,26]. In our previous work, it has been proved that ZnO can be loaded on the rods of PAL easily and successfully, and the prepared ZnO/PAL revealed excellent antibacterial properties [27,28], which may be the better candidate of filler used to reinforce CS-based wound dressings (films) for its dual function. In addition to ZnO/PAL, the naturally occurring compounds glycyrrhizic acid (GA), derived from the roots and rhizomes of legume licorice, has many pharmacological activities and has been testified to be active against drug-resistant bacteria [29,30]. For example, Zhao et al. prepared an injectable low-molecular-weight hydrogel by dissolving GA in physiological phosphate buffered saline (PBS) and found that the growth of *S. aureus* can be completely inhibited by this hydrogel [31]. However, there are no systematic studies about the effect of GA on improving the antibacterial properties of CS-based films up to now.

Against the background above, the CS-based films were designed with GA and ZnO/PAL as nano-filler in this study. By regulating the concentration of ZnO/PAL nanorods, a range of films with diverse morphology, mechanical properties, physical properties were acquired and characterized, and then emphasis was placed on the antibacterial efficacy of this potential wound dressing, aimed to evaluate the synergistic effect of CS, GA and ZnO/PAL in inhibiting wound infections that are caused by pathogens during tissue damage.

## 2. Materials and Methods

### 2.1. Materials

Chitosan (CS, high molecular weight: 800 kDa, deacetylation degree: 85%), glycyrrhizic acid (GA), and glacial acetic acid were purchased from Xingcheng Biological Products Factory Co., Ltd. (Nantong, Jiangsu, China), Gansu Fanzhi Pharmaceutical Co., Ltd. (Lanzhou, Gansu, China), and Tianjin Kaitong Chemical Reagent Co., Ltd. (Tianjin, China), respectively. Glycerol (GL) was purchased from Rionlon Bohua Pharmaceutical Chemical Co., Ltd. (Tianjin, China). ZnO/palygorskite (ZnO/PAL) nanorods were synthesized in our previous work and the ratio of ZnO on ZnO/PAL composite is 15% [27]. The bacterial strains of Gram-negative *Escherichia coli* (*E. coli*), Gram-positive *Staphylococcus aureus* (*S. aureus*), and two strains of drug-resistance bacteria, extended-spectrum β-lactamases-producing *Escherichia coli* (ESBL—*E. coli*) and methicillin-resistant *Staphylococcus aureus* (MRSA), were provided by the Department of Clinical Laboratory Center of Lanzhou University Second Hospital (Lanzhou, China). Nutrient Agar Medium was purchased from Qingdao Hope Bio-Technology Co., Ltd. (Qingdao, China). All the solutions were prepared with deionized water.

### 2.2. Preparation of CG and CGZP Films

Half a gram of ZnO/PAL powders were added into 50 mL of deionized water, then magnetically stirred for 3 h followed by ultrasonic dispersion for 20 min. Glacial acetic acid (1%, *v*/*v*) and 0.015 g of GA powder (0.5 wt.% of the total mass of CS) were dissolved in 45.5 mL deionized water. Then, 0 wt.% (0 mL), 0.25 wt.% (0.75 mL), 0.5 wt.% (1.5 mL), 1.0 wt.% (3.0 mL), 2.5 wt.% (7.5 mL), 5.0 wt.% (15.0 mL) of ZnO/PAL suspension, 3 g of CS and the corresponding volume of deionized water were added in the GA solution. The mixed solution was stirred at ambient temperature for 6 h. Thereafter, 40 wt.% GL aqueous solution was added into the mixture for another 1 h to obtain homogeneous solutions. The film-forming solutions were spread out to the sterile Petri dishes and air-dried at room temperature, and the finally formed films were coded as CG, CGZP−0.25%, CGZP−0.5%, CGZP−1.0%, CGZP−2.5%, CGZP−5.0%, respectively.

### 2.3. Characterizations of CG and CGZP Films

The cross-section morphology of CG and CGZP nanocomposite films was studied by high-resolution scanning electron microscopy (SEM, JSM-6701F, JEOL, Tokyo, Japan) with an accelerating voltage of 5 kV and a working distance of 8 mm at a magnification of 10,000. Attenuated Total Reflectance Fourier Transform Infrared Spectrometer (ATR-FTIR) of the films were recorded on the FTIR Spectrometer (FTIR-Nicolet iS20, Thermo Fisher, Wilmington, DE, USA) from 4000–500 cm^−1^ at a resolution of 4 cm^−1^. X-ray diffraction (XRD) of the films were recorded on X’Pert PRO powder X-ray diffractometer (PAN analytical, Almelo, The Netherlands) at a step size of 0.167° from 2θ = 5°–35° with Ni-filtered Cu Kα (*n* = 0.1541 nm) radiation. Thermogravimetric (TG) analysis of the films was recorded on STA 8000/Frontier Instrument (Perkin-Elmer Co., Groningen, The Netherlands) under a nitrogen (N_2_) atmosphere from 30 to 800 °C with a heating rate of 10 °C/min. For the furnace and sample stage, the flow rates of N_2_ purge gas were 20 and 40 mL/min, respectively.

### 2.4. Mechanical Properties of CG and CGZP Films

The New SANS versatile tension testing machine (CMT4304, SANS Test Machine Co. Ltd., Shenzhen, China) was employed for the measurement of the tensile strength (TS) and elongation at break (EB) of the films. Before the measurement, CG and CGZP film samples (80 mm × 10 mm) were placed in a container containing a saturated solution of Mg(NO_3_)_2_ (53% relative humidity) for at least 72 h. Tests were operated in tensile mode with an initial distance and crosshead speed set at 40 mm and 10 mm/min, respectively. At least five measurements were taken for each specimen.

### 2.5. Physicochemical Properties of CG and CGZP Films

The water content (*WC*), swelling degree (*SD*), and water solubility (*WS*) of CG and CGZP films were determined gravimetrically. The initial weight (*W*_1_) of square-shaped films (2 cm × 2 cm) was weighed and the dry weight (*W*_2_) of the films was weighed after placing them in a 70 °C oven for 24 h to remove the surface adsorbed water. Then immersed the dried films in deionized water completely at room temperature for another 24 h, stripped away the surface water of the swollen films with filter papers and weighed (*W*_3_). Finally, the swollen films were re-dried in an oven at 70 °C for 24 h and weighed (*W*_4_). The *WC*, *SD*, and *WS* were calculated by the following Equations (1)–(3) [32].
*WC* = (*W*_1_ − *W*_2_) × 100/*W*_1_,(1)
*SD* = (*W*_3_ − *W*_2_) × 100/*W*_2_,(2)
*WS* = (*W*_2_ − *W*_4_) × 100/*W*_2_,(3)

The water contact angle (*WCA*) with the surface of CS-based films was determined using the sessile drop method (OCA20, Dataphysics, Filderstadt, Germany). Water droplets with a volume of about 5 μL were deposited on the horizontal stage fitted with films (3 cm × 6 cm) to assess the hydrophobicity on the surface of the films at room temperature. *WCA* was measured on both sides of water droplets randomly for 5 measurements [33].

In order to determine the pH of CG and CGZP films, the 1 cm × 1 cm samples were mixed with 3 mL of normal saline solution (0.9%) at ambient temperature for 24 h. The pH of the mixed solutions was measured with a pH meter (Mettler Toledo, the United States), and the test was repeated three times to calculate average values [34].

### 2.6. Light Transmittance and Opacity of CG and CGZP Films

The light transmittance of CG and CGZP films in the range of 200–800 nm were recorded via UV 1900 UV-vis spectrophotometer (Shimadzu Corp., Tokyo, Japan). Before the test, the film samples were cut into 10 mm × 40 mm strips. Films with a greater opacity value are less transparent. The opacity values of the films were calculated by Equation (4) [35].
Opacity value = −log*T*_600_/*x*,(4)
where *T*_600_ is the transmittance at 600 nm, and *x* is the thickness (mm) of the films which were measured using an electronic outside micrometer.

### 2.7. Antibacterial Activity of CG and CGZP Films

The antibacterial activity of CG and CGZP films against two typical pathogenic bacteria (*E. coli* and *S. aureus*) and two drug-resistance bacteria (ESBL—*E. coli* and MRSA) were assessed by the agar diffusion method and colony counting method [36,37]. All these pathogenic strains were cultured overnight at 37 °C and serially diluted to 1.5 × 10^5^ CFU/mL. In the agar diffusion method, each film precursor solution (30 μL) was inoculated on the surface of agar media coated with 100 μL of bacterial suspensions. After 16~20 h of the incubation period (37 °C, 5% CO_2_), the diameters of inhibition zones around the discs were measured in mm and recorded as the antibacterial effect of CG and CGZP films. While the colony counting method was performed by incubating various film samples (3 cm × 3 cm) with 10 mL of each bacterial suspension (1.5 × 10^5^ CFU/mL) at 37 °C under shaking conditions (200 rpm/min) for 30 min. At the end of the incubation of films and bacterial suspensions, 100 μL of the culture was uniformly spread on the surface of the nutrient agar and further incubated for 16~20 h in a carbon dioxide incubator (37 °C, 5% CO_2_). The plates only with 100 μL bacterial suspensions were set as the controls. The percentage (%) of antibacterial effectiveness of CG and CGZP films was calculated as Equation (5). All the tests were made in three replicates for each film.
Antibacterial effectiveness (%) = (*A*
*− B*) × 100/*A*,(5)
where *A* represents the survived colonies on the agar plate after treating with bacterial suspensions, and *B* represents the survived colonies after treating with films soaked in bacterial suspensions.

### 2.8. Hemolysis Experiment of CG and CGZP Films

The hemolysis assay was conducted using fresh blood which was obtained from a healthy donor. The anticoagulant blood preliminary treated with EDTA was centrifuged at 2000 rpm for 10 min to obtain red blood cells (RBCs). The separated RBCs were washed 5 times with phosphate buffer saline (PBS, pH 7.4) and then diluted to a concentration of 5% (*v*/*v*) with PBS. Subsequently, the films (1 cm × 1 cm) were introduced into microcentrifuge tubes containing 200 μL of RBCs (5%, *v*/*v*) with 800 μL of PBS and incubated for 2 h at 37 °C. RBCs incubated with PBS and water were served as negative and positive controls. Following incubation, the hemolysis percentage (*HP*) was measured spectrophotometrically by reading the absorbance of the supernatant at 540 nm by a microplate reader (Infinite 200 Pro, Tecan, Männedorf, Switzerland). The *HP* (%) was calculated using Equation (6) [6].
*HP* (%) = (*OD_test_* − *OD_neg_*) × 100/(*OD_pos_* − *OD_neg_*)(6)
where *OD_test_*, *OD_pos_*, and *OD_neg_* is the absorbance value of RBCs solution following treatment with films, water and PBS, respectively.

### 2.9. Statistical Analysis

The statistical analysis of the results using the SPSS 25 software (SPSS INC., IBM, New York, NY, USA), model was validated through the analysis of variance (ANOVA). The differences between the average values were evaluated by Duncan’s test. The significance level was *p* < 0.05. The results were presented as the mean ± SD (standard deviation) (*n* = 3).

## 3. Results

### 3.1. Characterizations of CG and CGZP Films

Figure 1a shows the simulation process of film preparation containing physical and chemical interactions, and these interactions were systematically characterized by ATR-FTIR, XRD, SEM and TG as below. As shown in Figure 1b, the interactions between functional groups of CS, GA, and ZnO/PAL were analyzed on the basis of ATR-FTIR results. As for CG film, the broad absorption bands around 3500–3000 cm^−1^ were assigned to the O–H and N–H stretching vibration (–OH, –NH_2_, also between –OH and –NH_2_), the absorption peaks at 2882 cm^−1^ were assigned to the C–H symmetric stretching vibrations of –CH_2_ groups [38,39], at 1632 cm^−1^, 1551 cm^−1^ and 1332 cm^−1^ can be attributed to the C=O stretching of amide I, N–H bending of amide II and C–N stretching of amide III, respectively [40]. Moreover, the absorption peak at 1408 cm^−1^ can be attributed to the bending stretching of –CH_2_ [41], at 1150 cm^−1^ were assigned to the symmetric stretching of C–O–C bond, and at 1028 cm^−1^ were assigned to the C–O stretching vibrations [42,43,44]. In addition, the C=O stretching vibrations of GA can also be observed at 2943 cm^−1^ [45]. With the introduction of ZnO/PAL nanorods (2.5 wt.% or 5.0 wt.%) into the CG film, the peak at 1408 cm^−1^ shifted to 1402 cm^−1^, and the peak at 1028 cm^−1^ shifted to 1021 cm^−1^, indicating the interaction of CS, GA and ZnO/PAL nanorods. However, compared with CG, few changes were observed in CGZP−0.25%, CGZP−0.5% and CGZP−1.0% films, which is probably due to the low amount of ZnO/PAL added.

As shown in Figure 1c, the crystal structure characteristics of CG and CGZP films were analyzed by XRD. The diffraction peaks at 8.12° and 11.09° of CG film signifying the hydrated crystalline structure of CS bound with water molecules, while the diffraction peaks at 17.82° and 20.9° ascribe to the regular lattice and semicrystalline nature of CS [46,47,48]. It is noteworthy that the diffraction peaks of CGZP film at 8.12°, 11.09°, and 17.82° become flatter, and with the increase of ZnO/PAL nanorods, they finally disappear when the concentration of ZnO/PAL up to 5.0 wt.%. The observed results represented that the crystallinity of the films weakened after the combination of ZnO/PAL, as a result of the disruption of the original spatial structure of CS, which may be put down to newly generated intermolecular interactions. Moreover, the peaks associated with ZnO/PAL in the polymer matrix were barely visible. For one thing, this can be due to the structural deformation of PAL after calcination at 400 °C [27,49,50], and for another thing, the low concentration of ZnO/PAL nanorods in CS matrix would also affect the XRD results, which are similar to the reports of El-Sayed et al. [51] and Lin et al. [52].

SEM was used to analyze the distribution uniformity of GA and ZnO/PAL nanorods in CS matrix, which is depicted in Figure 1d–i. The smooth fracture surface morphology of CG film in Figure 1d indicated the well compatibility between CS and GA, and with the incorporation of ZnO/PAL, the unique nanorods can be observed in the form of not only rods but also bundles, revealing the presence ZnO/PAL nanorods in CG matrix, but did not achieve the desirable blending, which can be attributed to the intermolecular interaction between ZnO/PAL nanorods and the matrix [53]. The dispersion of clay fillers in polymer matrix plays a crucial effect on the physicochemical properties of the films, as confirmed by Mohamed et al. [54]. It can be found that with the increased amount of nanorods, the folds of the films increased accordingly, some cracks and aggregations appeared at the same time, which may negatively influence the mechanical properties for its stress defects (the decreased EB at mechanical properties).

TG and DTG analysis exhibited the thermal stability of CG and CGZP nanocomposite films. As shown in Figure 2, the thermal decomposition of CG and CGZP films all occurred in a four-stage process. The first phase of weight loss occurred around 30–105 °C with the peak temperature of 64 °C can be attributed to the evaporation of surface adsorbed water, and also the water molecules bound mainly in –NH_2_ and –OH of the CS [55]. The second and third phases of weight loss (105–160 °C and 160–238 °C) corresponds to the removal of acetic acid, evaporation of glycerol, respectively [56]. The final stage of weight loss starts at 238 °C and goes up to 400 °C is due to the decomposition of CS and GA because of its depolymerization process [57]. After the last stage of thermal decomposition, the remnant percentage of the CGZP−5.0% film is around 28% at 800 °C and 23% for CG. The higher residual of CGZP−5.0% is mainly for the presence of ZnO/PAL nanorods, which endows it with better thermal stability.

### 3.2. Mechanical Properties of CG and CGZP Films

Excellent mechanical property is the fundamental feature of ideal wound dressings to prevent wounds from trauma during the process of healing [58]. Thus, the mechanical properties of the films were investigated in detail, including tensile strength (TS) and elongation at break (EB). As shown in Figure 3, the pure CG film has a TS of 23.5 MPa and EB of 44.8%, though the EB is enough, the low TS of CG limits its end-up applications. Compared with CG, the TS and EB showed an upward and downward trend with the incorporation of ZnO/PAL nanorods, respectively, where the maximum value of TS (37.5 MPa) and minimum value of EB (30.0%) were attained in CGZP−5.0% film, indicating statistically significant difference (*p* < 0.05). So CGZP−5.0% film can be the better choice for wound dressing than CG film. The results may be due to the strong interaction between ZnO/PAL nanorods, GA, and CS. Specifically speaking, the highest TS of CGZP−5.0% film might be owing to the most powerful electrostatic interaction and hydrogen bonding between the –NH_2_ and –OH groups of CS, –OH and –COOH of GA and silanol groups of PAL [59]. Moreover, the high aspect ratio of ZnO/PAL (as the interface coupling agent of CS and GA) can afford intermolecular bonds with higher bond energy, which is conducive to the rotation of molecular chains, also contributes to the improvement of rigidity of CGZP films [55,60,61]. On the other hand, the defects caused by the excessive introduction of nanofiller (confirmed by SEM in Figure 1) are the main culprit for the reduction of EB.

### 3.3. Physicochemical Properties of CG and CGZP Films

The physical properties of the nanocomposite films were summarized in Table 1, including water content (*WC*), water solubility (*WS*) and swelling degree (*SD*). High humidity environments are generally needed for wound dressings to encourage the concrescence of the wounds, therefore it is of vital significance to identify the *WC* and *WS* of as-prepared films [62]. As can be observed from Table 1, the addition of ZnO/PAL nanorods helped to increase the *WC* and *SD* of the films, while decreasing the *WS* value. Compared with CG, the *WC* of CGZP−5.0% film increased from 11.21% to 17.13% (a 52.81% increase, *p* < 0.05), the *SD* increased from 325.94% to 472.49% (a 44.96% increase, *p* < 0.05), and the *WS* decreased from 28.15% to 21.58% (a 23.34% decrease, *p* < 0.05). Tabatabaei et al. reported that *WC* is in connection with empty spaces of the nanocomposite structure [63]. The lower the *WS,* the more beneficial to the applications as biodegradable films; the higher the *SD*, the easier the films to absorb wound exudates to prevent wounds from infection, thus high *SD* is one of the most considerable characteristics of the wound dressings [64]. The results were in accordance with the reports by Gasti et al. [65]. In addition, Figure 4a shows the water contact angle (*WCA*) of CG and CGZP nanocomposite films, applying for the assessment of surfaces wettability (hydrophilicity). The *WCA* of the films was in the range of 103.5°–115.5°. It is well known that CS is poorly soluble in neutral water, and apparently *WCA* of CG film is the highest among all the films (*p* < 0.05). The hydrophobicity of films decreased after incorporating with ZnO/PAL nanorods, which for one thing, may be involved with the combination of highly hydrophilic ZnO/PAL nanorods, and for another thing, maybe due to the decrease of pH of the films. As depicted in Figure 4b, the pH of the films falls in the range of 5~6, which is in accordance with some commercially available wound dressing products [66]. Compared with CG film, an obvious reduction was presented in CGZP−5.0% film (*p* < 0.05), which contributes to the protonation of CS (decrease of *WCA*), also furnishes an appropriate slightly acidic environment to promote cell proliferation [67].

### 3.4. Light Transmittance and Opacity of CG and CGZP Films

The optical property of the wound dressings is a significant element in real-time monitoring of the change of wounds [68]. The light transmittance spectra of CG and CGZP films are presented in Figure 5, where CG film has the highest light transmittance of 91% at 800 nm, and as predicted, the light transmittance of the films decreased slightly with the increased content of ZnO/PAL nanorods, and finally dropped to 86% in CGZP−5.0% film. A higher concentration of ZnO/PAL increased the opacity of CGZP films, which might be attributed to the existence of ZnO/PAL that may block light from traversing the film in the form of scattering, especially when nanorods are evenly decentralized in the polymer matrix [69]. Furthermore, the thickness may also affect the opacity of the films, although the differences between them are not significant (*p* > 0.05), as revealed in Table 2.

### 3.5. Antibacterial Activity of CG and CGZP Films

Antibacterial activity has been demonstrated to be a superiority of excellent bioactive materials, and its facilitating role in wound healing has also been proved [70,71]. Therefore, a sequence of experiments on the antibacterial activity of the films were carried out. Primarily, the agar diffusion method was conducted to qualitatively evaluate the antibacterial activity of the films as presented in Figure 6a and Table 3. It can be found that CG has the ability to inhibit bacterial growth, but the bacteriostatic activity of CG is insufficient if used as wound dressings. The addition of ZnO/PAL helped to improve the deficiency (*p* < 0.05), and the higher the concentration of ZnO/PAL nanorods, the better the antibacterial effect of the films, though ESBL—*E. coli* and MRSA have smaller inhibitory zone diameters than those of *E. coli* and *S. aureus*. Posteriorly, the colony counting method was used to quantitatively evaluate the antibacterial activity of CG and CGZP films (Figure 6b, c). Similar to agar diffusing, the total number of bacterial colonies decreased obviously when they met with CG, which was able to inhibit the growth of 90.5% of *E. coli*, 95.8% of *S. aureus*, 95.9% of ESBL—*E. coli*, 94.4% of MRSA, respectively. Though ZnO/PAL did not significantly affect the activity of ESBL—*E. coli* (*p* > 0.05), it still showed excellent antibacterial performance. The inhibiting trend continued with the increased amount of ZnO/PAL, and hardly any bacterial colonies were found alive when the content of ZnO/PAL increased to 5.0 wt.% (able to inhibit the growth of 99.5% of *E. coli*, 99.8% of *S. aureus*, 99.6% of ESBL—*E. coli*, 99.8% of MRSA, respectively). The remarkable antibacterial properties can be attributed to the synergistic effect of CS, GA and ZnO/PAL [72,73]. Yilmaz et al. reported that the polycationic structure of CS is the main reason for its antibacterial activity, CS can interact with the negatively charged bacterial cell membrane, leading to a conspicuous increase in permeability of cell wall, and then attach to DNA leading to inhibition of DNA replication, followed by bacterial death [74,75]. Kim et al. confirmed that triterpenoid aglycon of GA has the ability to inhibit the synthesis of DNA, RNA and protein, thus causing bacterial death [76]. The antibacterial property of ZnO/PAL is mainly attributed to ZnO loaded in PAL nanorods. ZnO can slowly release Zn^2+^ in the film precursor solution, which may destroy the cell walls of bacteria and make it easier to cling to the surface of bacterial cell membranes, and therefore restrain the bacterial from growing [21]. Additionally, the released Zn^2+^ can bind to the negatively charged surface of the bacterial cell membrane by electrostatic attraction, breaking the charge balance on the surface and causing bacterial lysis [77]. Moreover, the rod-like PAL may also cause mechanical damage to the bacteria. Contextually, the synergy between CS, GA and ZnO/PAL provided CGZP well-pleasing multi-antibacterial properties, making them worthy candidates for wound healing application.

### 3.6. Hemocompatibility of CG and CGZP Films

In order to estimate the hemocompatibility of the films, the hemolysis assay was conducted by determining the amount of hemoglobin released from RBCs when the blood contacts with films [78]. Materials can be sorted into three categories contingents on hemolysis percentage (*HP**)*: *HP* < 2%, nonhemolytic; 2% ≤ *HP* < 5%, slightly hemolytic; *HP* ≥ 5%, hemolytic [79]. As shown in Figure 7a, after 2 h of incubation, the *HP* of all the films were lower than 2% and showed a significant difference (*p* < 0.05), indicating all the films are nonhemolytic materials, which can also be proved by the color of supernatants as presented in Figure 7b. Apparently, there is no particular distinction in color between CG and CGZP films, whereas the negative (PBS) and positive control (H_2_O) groups are transparent and red. Analogously, the microscope images (Figure 7c) of CGZP−5.0% film displayed no change in the morphology of erythrocytes after incubation with film, while no erythrocytes were found in the positive control group, suggesting the excellent hemocompatibility of the films. All the results above suggested the safety and reliability of the films for wound dressings.

As one of the potential wound dressings, in vivo evaluation in this study is far from sufficient. To further validate the functionality and performance of this wound healing system, we propose carrying out some in vivo trials in the following work to evaluate whether it is suitable for the treatment of actual wound infections, such as observing the change of wound area over time through animal experiment, evaluating its antibacterial performance and biocompatibility. If it is confirmed to be clinically applicable, the comparison between this wound dressing (CGZP) and current clinically used ones would also need to conduct to further explore its application value.

## 4. Conclusions

A series of CG and CGZP films have been fabricated by the introduction of GA and ZnO/PAL nanorods (0.25 wt.%–5.0 wt.%) into CS matrix for the potential application in wound healing. It was the interactions between –OH and –NH_2_ of CS, –OH and –COOH of GA, and silanol groups of PAL that contribute to the dispersion uniformity of nanofillers in CS matrix, which were characterized and proved by SEM, ATR-FTIR and XRD. Furthermore, compared with CG, the improved mechanical properties of CGZP helps raise higher tension, the enhanced physical properties (especially water content and swelling degree) helps provide a high humidity environment and absorb wound exudates, while the excellent light transmittance helps the real-time monitoring of the change in wounds, which all endow CGZP with more intentional performance as wound dressings. Moreover, it was confirmed by the inhibition zone test and colony counting method that the films, especially CGZP−5.0%, have excellent antibacterial performance against familiar pathogenic bacteria and even drug-resistant bacteria, which can be attributed to the synergic antibacterial activity of CS, GA and ZnO/PAL. In addition, the outstanding hemocompatibility of the films has also been affirmed by the hemolysis study. Therefore, the combination of CS, GA, and ZnO/PAL is expected to be deployed in the preparation of wound-care products.

## Figures and Tables

**Figure 1 polymers-13-03878-f001:**
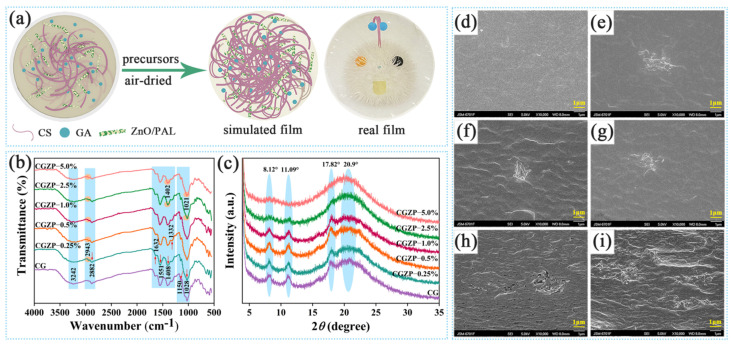
The simulation process of films (the digital photo of the real film was taken by overlaying the film on a photo of cartoon character we drew by ourselves) (**a**), ATR-FTIR spectra of CG and CGZP films (**b**), XRD patterns of CG and CGZP films (**c**), SEM micrograph of the cross-section of CG (**d**), CGZP−0.25% (**e**), CGZP−0.5% (**f**), CGZP−1.0% (**g**), CGZP−2.5% (**h**), and CGZP−5.0% (**i**) films with a magnification of 10,000.

**Figure 2 polymers-13-03878-f002:**
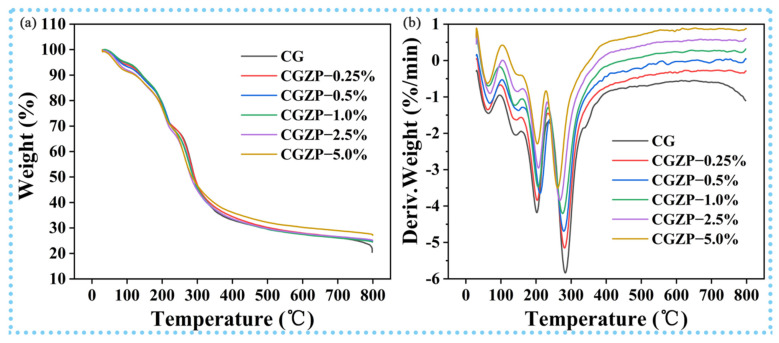
TG curves (**a**) and DTG curves (**b**) of CG and CGZP films.

**Figure 3 polymers-13-03878-f003:**
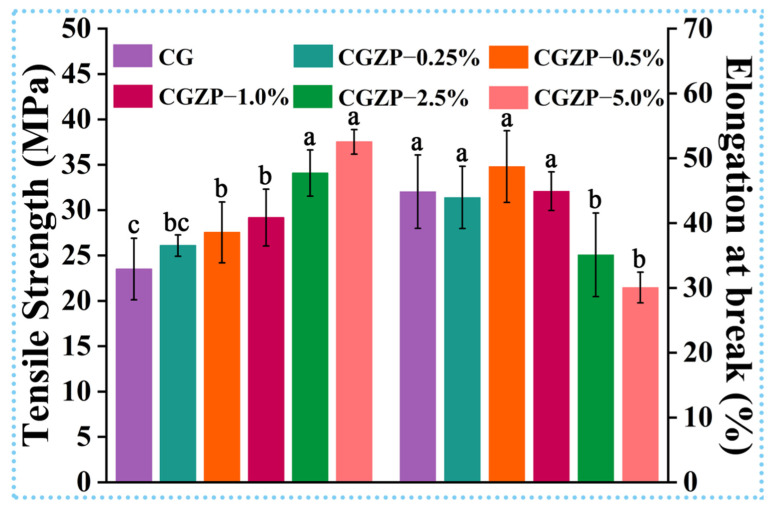
Tensile strength and Elongation at break of CG and CGZP films. Different letters in the columns indicate significant difference (*p* < 0.05).

**Figure 4 polymers-13-03878-f004:**
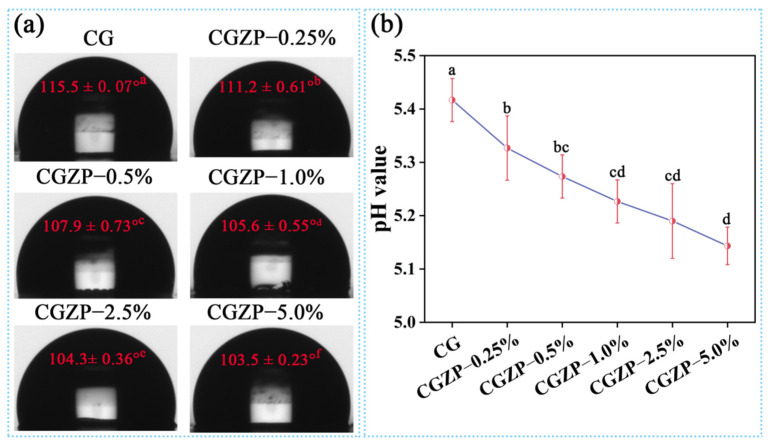
*WCA* (**a**) and pH (**b**) value of CG and CGZP films. Different letters in each figure indicate significant differences (*p* < 0.05).

**Figure 5 polymers-13-03878-f005:**
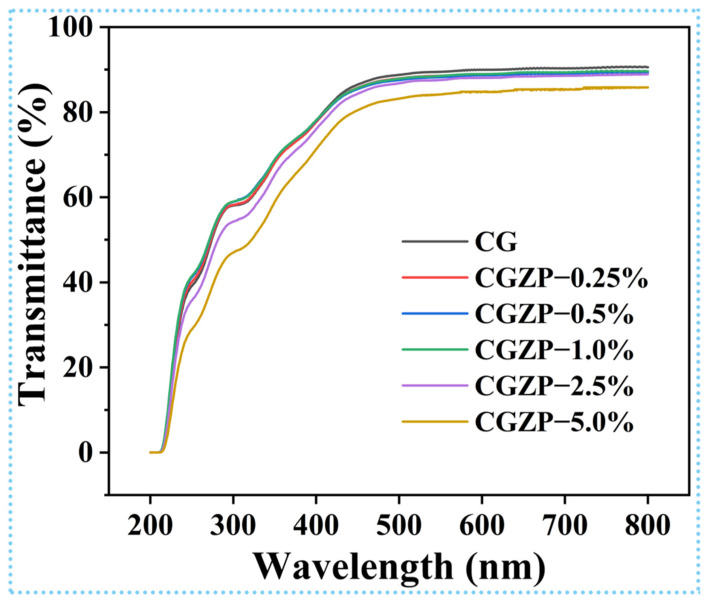
The light transmittance spectra of the films.

**Figure 6 polymers-13-03878-f006:**
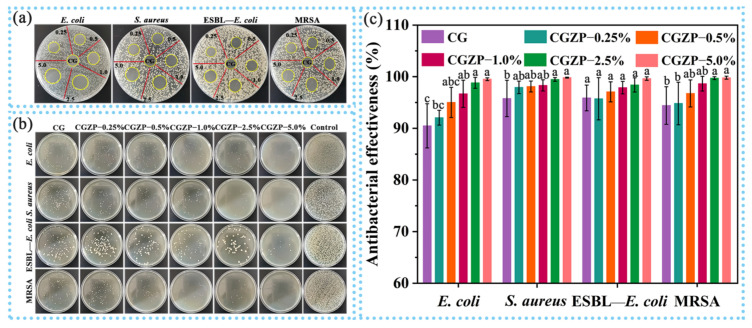
Photographs of inhibitory zones (**a**) and bacterial colonies (**b**) of CG and CGZP films against *E. coli*, *S. aureus*, ESBL—*E. coli*, and MRSA, (**c**) quantitative statistics of antibacterial effectiveness of colony counting method, and different letters in the columns indicate significant differences (*p* < 0.05).

**Figure 7 polymers-13-03878-f007:**
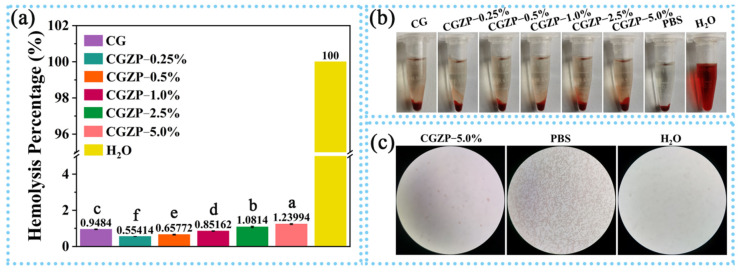
(**a**) Hemolysis Percentage of CG and CGZP films, and different letters in the columns indicate significant difference (*p* < 0.05), (**b**) photographs of centrifuge tubes by disposing of RBCs with different films, PBS, and water, (**c**) microscope images of CGZP−5.0% film and the controls treated with RBCs.

**Table 1 polymers-13-03878-t001:** Thickness, water content (*WC*), water solubility (*WS*) and swelling degree (*SD*) of the films.

Samples	*WC* (%)	*WS* (%)	*SD* (%)
CG	11.21 ± 0.91 ^e^	28.15 ± 0.92 ^a^	325.94 ± 1.77 ^f^
CGZP−0.25%	12.92 ± 0.89 ^d^	27.35 ± 1.58 ^a^	384.78 ± 2.07 ^e^
CGZP−0.5%	13.78 ± 0.27 ^c,d^	26.67 ± 0.92 ^a,b^	394.57 ± 3.90 ^d^
CGZP−1.0%	14.87 ± 0.66 ^b,c^	24.39 ± 0.90 ^b,c^	415.39 ± 3.79 ^c^
CGZP−2.5%	15.75 ± 0.53 ^a,b^	22.12 ± 0.41 ^c,d^	433.54 ± 4.43 ^b^
CGZP−5.0%	17.13 ± 0.39 ^a^	21.58 ± 1.19 ^d^	472.49 ± 3.32 ^a^

Different superscript letters in the same column demonstrate significant difference between values (*p* < 0.05).

**Table 2 polymers-13-03878-t002:** Transmittance (%) at different wavelengths (200–800 nm), thickness and opacity for CG and CGZP films.

Samples	Transmittance (%)							Thickness (mm)	Opacity
	200 nm	300 nm	400 nm	500 nm	600 nm	700 nm	800 nm		
CG	0.02	58.05	78.01	88.76	89.94	90.31	90.54	0.0548 ± 0.0018 ^a^	0.84 ± 0.03 ^e^
CGZP−0.25%	0.02	58.20	77.67	87.93	88.96	89.34	89.51	0.0537 ± 0.0016 ^a,b^	0.95 ± 0.03 ^d^
CGZP−0.5%	0.02	58.93	77.99	87.55	88.68	89.08	89.38	0.0533 ± 0.0018 ^a,b^	0.98 ± 0.03 ^c,d^
CGZP−1.0%	0.01	58.98	78.12	87.84	88.93	89.38	89.59	0.0529 ± 0.0017 ^b^	0.96 ± 0.03 ^c^
CGZP−2.5%	0.02	54.26	76.02	86.80	88.07	88.54	88.91	0.0527 ± 0.0013 ^b^	1.05 ± 0.03 ^b^
CGZP−5.0%	0.02	54.46	74.69	84.00	85.26	85.74	86.08	0.0533 ± 0.0018 ^a,b^	1.30 ± 0.04 ^a^

Different superscript letters in the same column demonstrate significant differences between values (*p* < 0.05).

**Table 3 polymers-13-03878-t003:** Inhibitory zone diameters of the films against different bacteria.

Inhibitory Zones in Diameter (mm)	*E. coli*	*S. aureus*	ESBL—*E. coli*	MRSA
CG	13.44 ± 0.53 ^b,c^	13.70 ± 0.32 ^b^	12.54 ± 0.34 ^b,c^	12.36 ± 0.31 ^c^
CGZP−0.25%	12.64 ± 0.86 ^c^	13.13 ± 0.45 ^b^	11.72 ± 0.64 ^c^	12.06 ± 0.55 ^c^
CGZP−0.5%	13.36 ± 0.31 ^b,c^	13.55 ± 0.38 ^b^	12.08 ± 0.52 ^b,c^	12.30 ± 0.44 ^c^
CGZP−1.0%	13.50 ± 0.32 ^b,c^	13.73 ± 0.15 ^b^	12.30 ± 0.44 ^a,b^	12.62 ± 0.53 ^b,c^
CGZP−2.5%	14.14 ± 0.51 ^a,b^	14.68 ± 0.38 ^a^	12.84 ± 0.72 ^a,b^	13.12 ± 0.57 ^a,b^
CGZP−5.0%	15.04 ± 1.07 ^a^	14.50 ± 0.42 ^a^	13.28 ± 0.31 ^a^	13.64 ± 0.29 ^a^

Different superscript letters in the same column demonstrate significant differences between values (*p* < 0.05).

## Data Availability

Not applicable.

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
