# Peer review of "Synergistic Effect of Glycyrrhizic Acid and ZnO/Palygorskite on Improving Chitosan-Based Films and Their Potential Application in Wound Healing"

_polymers, 2021, doi:10.3390/polym13223878_

Round 1
Reviewer 1 Report
The review "Synergistic effect of glycyrrhizic acid and ZnO/palygorskite on improving chitosan-based films and their potential application in wound healing" describes the obtaining of a new type of composite with potential uses for wound dressing. The subject of manuscript is worthy of investigation and it can be published after following corrections.
English language must be polished (e.g. row 133 “to detect their mass”; row 192 “Bisic characterization”; row 196 “were dissected”; row 221 “dispersionuniformity” etc). Authors are advised to seek help from a native English speaker.
Please increase the size of figures for better readability (figure 1, 2 – use the full width of page). At SEM especially put only 2 pictures on a row and use the available lateral space. This will improve a lot the readability, the possibility of other scientists to see the results, which will increase the impact of the article.
For previous results on antibacterial activity of ZnO/Chitosan composite against E.coli and S. aureus please refer to and cite doi: 10.3390/foods9121801; 10.1016/j.ijpharm.2013.11.035. For antimicrobial chitosan formulation doi: 10.2174/138920101602150112151157
Authors should provide a ratio for ZnO/PAL composite (e.g. 10% ZnO) – see further the XRD discussion. The XRD patterns are modified for at least two samples (0.25% and 1%) for which in the region 28-35 degree the color of line was changed. Why stop the XRD at 35 degree? Please give the full pattern, up to at least 60 degree. Why there are no peaks from ZnO/PAL (at least for 2.5 and 5% samples the inorganic phase should be detectable)? The phrase from rows 213-214 must be rewritten as there are no peaks in XRD due to amorphous substances.
In TGA section, at row 242 word glacial must be eliminated. There are some residual acetic acid molecules that are eliminated, but not "glacial acetic acid". Furthermore, the glycerol is also eliminated by evaporation not by decomposition (in the ref 50 is mentioned the elimination). Please specify the atmosphere condition from TG analysis. It is in air or in inert gas (which?), because it will affect the type of processes (decompositions / oxidations). As it is right now, if the inorganic phase is 5% why the residual mass is around 28%?
This reviewer wants to know how was film thickness measured?
The rows 309-311 – the opacity of CGZP films is increasing with the increasing ZnO/PAL %. So the correct sentence will be “The increased opacity....” As authors have the values of transmittance and film thickness, they can provide additional information for increasing the manuscript scientific impact by calculating the opacity values (see 10.3390/pharmaceutics13071020 for the quick method).
Author Response
Comments and suggestions: The review "Synergistic effect of glycyrrhizic acid and ZnO/palygorskite on improving chitosan-based films and their potential application in wound healing" describes the obtaining of a new type of composite with potential uses for wound dressing. The subject of the manuscript is worthy of investigation and it can be published after following corrections.
Question (1): English language must be polished (e.g. row 133 “to detect their mass”; row 192 “Bisic characterization”; row 196 “were dissected”; row 221 “dispersion uniformity” etc). Authors are advised to seek help from a native English speaker.
Author’s reply: Thank you very much for your suggestions. We have modified the sentence you mentioned, revised the entire manuscript carefully and improved our English language. All the changes are marked using the “Track Changes” function in the revised manuscript.
Question (2): Please increase the size of figures for better readability (figure 1, 2 – use the full width of page). At SEM especially put only 2 pictures on a row and use the available lateral space. This will improve a lot the readability, the possibility of other scientists to see the results, which will increase the impact of the article.
Author’s reply: Thank you very much for your suggestions. We have increased the size of Figure 1 and Figure 2 (use the full width of the page), also put 2 pictures on a row in SEM to improve the readability of the results.
Question (3): For previous results on antibacterial activity of ZnO/Chitosan composite against E. coli and S. aureus please refer to and cite doi: 10.3390/foods9121801; 10.1016/j.ijpharm.2013.11.035. For antimicrobial chitosan formulation doi: 10.2174/138920101602150112151157
Author’s reply: Thank you very much for your suggestions. We have carefully read the literature you mentioned and quoted them to the revised manuscript to increase the credibility of the results.
Question (4): Authors should provide a ratio for ZnO/PAL composite (e.g. 10% ZnO) – see further the XRD discussion. The XRD patterns are modified for at least two samples (0.25% and 1%) for which in the region 28-35 degree the color of line was changed. Why stop the XRD at 35 degree? Please give the full pattern, up to at least 60 degree. Why there are no peaks from ZnO/PAL (at least for 2.5 and 5% samples the inorganic phase should be detectable)? The phrase from rows 213-214 must be rewritten as there are no peaks in XRD due to amorphous substances.
Author’s reply: Thank you very much for your suggestions and questions, the ratio of ZnO on ZnO/PAL composite (15% ZnO) has been supplemented in "2.1. Materials" section. And thanks again for the error you pointed out, the different color in the XRD patterns of CGZP-0.25%, CGZP-1.0% was due to a drawing error (we did not notice that before), rather than a modification of original data, and we have corrected their color. Through our previous researches and literature reviews, we found that the XRD diffraction peaks of CS almost appear under 2θ = 35°, so we stopped collecting XRD data at 2θ = 35° [R1–R3]. And unfortunately, my city is seriously affected by the Covid-19 at present, all experiments outside the school were broken off for safety, so we can not support another XRD testing within 10 days. I do apologize for any inconvenience this may cause. The peaks associated with ZnO/PAL in the polymer matrix were barely visible, for one thing, this can be due to the structural deformation of PAL after calcination at 400 ℃ [R4–R6], and for another thing, the low concentration of ZnO/PAL nanorods in CS matrix would also affect the XRD results, which were similar to the reports of El-Sayed et al [R7] and Lin et al [R8]. Besides, we have rewritten the phrase you mentioned in the revised manuscript.
Related references:
[R1] Ding, J.; Hui, A.; Wang, W.; Yang, F.; Kang, Y.; Wang, A. Multifunctional palygorskite@ ZnO nanorods enhance simultaneously mechanical strength and antibacterial properties of chitosan-based film. Int. J. Biol. Macromol. 2021, 189, 668-677. https://doi.org/10.1016/j.ijbiomac.2021.08.107
[R2] Zhang, H;, Wang, W.; Ding, J.; Lu, Y.; Xu, J.; Wang, A. An upgraded and universal strategy to reinforce chitosan/polyvinylpyrrolidone film by incorporating active silica nanorods derived from natural palygorskite. Int. J. Biol. Macromol. 2020, 165, 1276-1285. https://doi.org/10.1016/j.ijbiomac.2020.09.241
[R3] Zeng, J.; Ren, X.; Zhu, S.; Gao, Y. Fabrication and characterization of an economical active packaging film based on chitosan incorporated with pomegranate peel. Int. J. Biol. Macromol. 2021, 192, 1160-1168. https://doi.org/10.1016/j.ijbiomac.2021.10.064
[R4] Hui, A.; Yan, R.; Wang, W.; Wang, Q.; Zhou, Y.; Wang, A. Incorporation of quaternary ammonium chitooligosaccharides on ZnO/palygorskite nanocomposites for enhancing antibacterial activities. Carbohydr. Polym. 2020, 247, 116685. https://doi.org/10.1016/j.carbpol.2020.116685
[R5] Huo, C; Yang, H. Synthesis and characterization of ZnO/palygorskite. Appl. Clay Sci. 2010, 50, 362-366. https://doi.org/10.1016/j.clay.2010.08.028
[R6] Rosendo, F.R.; Pinto, L.I.; de Lima, I.S.; Trigueiro, P.; Honório, L.M.D.C.; Fonseca, M.G.; Silva-Filho, E.C.; Ribeiro, A.B.; Furtini, M.B.; Osajima, J.A. Antimicrobial efficacy of building material based on ZnO/palygorskite against Gram-negative and Gram-positive bacteria. Appl. Clay Sci. 2020, 188, 105499. https://doi.org/10.1016/j.clay.2020.105499
[R7] El-Sayed, S.M.; El-Sayed, H.S.; Ibrahim, O.A.; Youssef, A.M. Rational design of chitosan/guar gum/zinc oxide bionanocomposites based on Roselle calyx extract for Ras cheese coating. Carbohydr. Polym. 2020, 239, 116234. https://doi.org/10.1016/j.carbpol.2020.116234
[R8] Lin, D.; Yang, Y.; Wang, J.; Yan, W.; Wu, Z.; Chen, H.; Zhang, Q.; Wu, D.; Qin, W.; Tu, Z. Preparation and characterization of TiO2-Ag loaded fish gelatin-chitosan antibacterial composite film for food packaging. Int. J. Biol. Macromol. 2020, 154, 123-133. https://doi.org/10.1016/j.ijbiomac.2020.03.070
Question (5): In TGA section, at row 242 word glacial must be eliminated. There are some residual acetic acid molecules that are eliminated, but not "glacial acetic acid". Furthermore, the glycerol is also eliminated by evaporation not by decomposition (in the ref 50 is mentioned the elimination). Please specify the atmosphere condition from TG analysis. It is in air or in inert gas (which?), because it will affect the type of processes (decompositions / oxidations). As it is right now, if the inorganic phase is 5% why the residual mass is around 28%?
Author’s reply: Thank you very much for your suggestions and questions. We have eliminated the word "glacial" and corrected the elimination process of glycerol, and specified the atmosphere condition (N2) of TG in "2.3. Characterizations of CG and CGZP films" section. Due to the decomposition of the polymers, the residual mass is around 28% in CGZP-5.0% film, as explained in the revised manuscript.
Question (6): This reviewer wants to know how was film thickness measured?
Author’s reply: Thank you very much for your question, the thickness of the films was measured using an electronic outside micrometer during the testing of mechanical properties, and we have adjusted the explanation of thickness to “2.6 Light transmittance and Opacity of CG and CGZP films” section.
Question (7): The rows 309-311 – the opacity of CGZP films is increasing with the increasing ZnO/PAL %. So the correct sentence will be “The increased opacity....” As authors have the values of transmittance and film thickness, they can provide additional information for increasing the manuscript scientific impact by calculating the opacity values (see 10.3390/pharmaceutics13071020 for the quick method).
Author’s reply: Thank you very much for your suggestions. We have corrected the sentence you mentioned, and the opacity of CG and CGZP films was calculated by the values of light transmittance and thickness, as illustrated in Table 2, and the opacity results are also discussed in the revised manuscript.
Reviewer 2 Report
The manuscript entitled “Synergistic effect of glycyrrhizic acid and ZnO/palygorskite on improving chitosan-based films and their potential application in wound healing”, by Wang et al have investigated synergistic effect of chitosan (CS), glycyrrhizic acid (GA) and ZnO/palygorskite (ZnO/PAL) as potential wound dressing systems when prepared as films. The research is well executed. However, there are some gaps in the current manuscript which if addressed, would significantly improve the usefulness of this work for the readers. Please note some comments listed below.
Comments to authors:
- Please discuss the effects of non-uniform compound distribution during film formation as seen by SEM analysis.
- Please discuss statistical testing results in all figure legends along with symbols on the graph (wherever applicable) to show significantly different groups.
- Please elaborate on the type of wounds this dressing can be used.
- Please include a section for the limitations of the films prepared.
- Please also include a section on future studies that are warranted to further validate the functionality and performance of this wound healing system (e.g. comparison with current clinically used controls, in vivo testing, stability/robustness testing, and so on)
- Please check and correct for grammatical and typographical errors throughout the manuscript.
Author Response
Comments and suggestions: The manuscript entitled “Synergistic effect of glycyrrhizic acid and ZnO/palygorskite on improving chitosan-based films and their potential application in wound healing”, by Wang et al have investigated synergistic effect of chitosan (CS), glycyrrhizic acid (GA) and ZnO/palygorskite (ZnO/PAL) as potential wound dressing systems when prepared as films. The research is well executed. However, there are some gaps in the current manuscript which if addressed, would significantly improve the usefulness of this work for the readers. Please note some comments listed below.
Question (1): Please discuss the effects of non-uniform compound distribution during film formation as seen by SEM analysis.
Author’s reply: Thank you very much for your suggestions. The non-uniform compound distribution during film formation may negatively influence the mechanical properties for its stress defects, and we added the relevant discussions in the revised manuscript.
Question (2): Please discuss statistical testing results in all figure legends along with symbols on the graph (wherever applicable) to show significantly different groups.
Author’s reply: Thank you very much for your suggestions. We have discussed statistical testing results in all figure legends and added symbols on the graphs to show significantly different groups in the revised manuscript.
Question (3): Please elaborate on the type of wounds this dressing can be used.
Author’s reply: Thank you very much for your suggestion. The possible type of wounds this dressing may treat has been elaborated on in the “1. introduction” section in the revised manuscript.
Question (4): Please include a section for the limitations of the films prepared.
Author’s reply: Thank you very much for your suggestion. The limitations of the films have been added in the revised manuscript.
Question (5): Please also include a section on future studies that are warranted to further validate the functionality and performance of this wound healing system (e.g. comparison with current clinically used controls, in vivo testing, stability/robustness testing, and so on).
Author’s reply: Thank you very much for your suggestions. The future studies that are warranted to further validate the functionality and performance of this wound healing system have been added to ”4. Conclusions” section in the revised manuscript.
Question (6): Please check and correct for grammatical and typographical errors throughout the manuscript.
Author’s reply: Thank you very much for your suggestions. We have carefully checked and corrected for grammatical and typographical errors throughout the manuscript, and all the changes are marked using the “Track Changes” function in the revised manuscript.
Round 2
Reviewer 1 Report
The authors have responded to my comments and have addressed all my concerns, substantially improving the manuscript, therefore, I suggest publishing the article in the current form.